# Including a Three-Party Meeting Using the Demand and Ability Protocol in an Interdisciplinary Pain Rehabilitation Programme for a Successful Return to Work Process

**DOI:** 10.3390/ijerph192416614

**Published:** 2022-12-10

**Authors:** Katarina Aili, Therese Hellman, Magnus Svartengren, Katarina Danielsson

**Affiliations:** 1Department of Medical Sciences, Occupational and Environmental Medicine, Uppsala University, 751 85 Uppsala, Sweden; 2Department of Health and Sport, School of Health and Welfare, Halmstad University, 301 18 Halmstad, Sweden; 3Department of Occupational and Environmental Medicine, Uppsala University Hospital, 751 85 Uppsala, Sweden; 4Department of Medical Sciences, Psychiatry, Uppsala University, 751 85 Uppsala, Sweden

**Keywords:** workplace intervention, rehabilitation, chronic pain, return to work

## Abstract

The Demand and Ability Protocol (DAP) is used in three-party meetings involving an employee, an employer, and a representative from the rehabilitation team. The aim of this study is to investigate the inclusion of an intervention using the DAP in an interdisciplinary pain rehabilitation programme (IPRP) compared to usual care. This non-randomised controlled trial included patients assigned to an IPRP in Sweden. The intervention group received a DAP intervention targeting their work situation in addition to the usual care provided by the IPRP. The control group received IPRP only. Outcome measures were collected from the Swedish Quality Registry for Pain Rehabilitation. Results demonstrated improvements in both groups regarding self-reported anxiety, depression and EQ5D. Sleep was improved in the intervention group but not in the control group. No statistical differences in outcomes were observed between the groups. In conclusion, adding the DAP intervention to IPRP seemed to have the potential to improve sleep among the patients, which may indicate an overall improvement regarding health outcomes from a longer perspective. The results were less clear, however, regarding the work-related outcomes of sickness absence and workability.

## 1. Introduction

Approximately 30–40% of the adult population in Sweden are likely to report chronic musculoskeletal pain (often defined as pain present for more than three months) over their lifespan. Chronic pain is more common among women, and the prevalence increases with higher age [1,2]. Low back pain is the most common chronic pain disorder, and it is one of the leading causes of years lived with disability [3]. In Sweden, musculoskeletal disorders are the second most common cause of sickness absence [4].

The clinical presentation of chronic pain is very heterogeneous, where some chronic pain conditions appear more complex, with more comorbidities and complex aetiology than others. Patients with complex chronic pain may be offered treatment by an interdisciplinary pain programme (IPRP). In Sweden, individuals with complex pain can receive interdisciplinary pain rehabilitation, either in primary care or in a specialist pain unit. Patients that are referred to a specialised unit have more severe pain or a more complex pain condition than those provided care at a primary care unit [5]. In Sweden, the IPRP is usually delivered by an interdisciplinary team of, e.g., physicians, physiotherapists, psychologists, social workers and occupational therapists. The rehabilitation programme includes group activities such as pain education, physical activity and activities that aim to promote a return to work (RTW). The programmes are often based on acceptance and commitment therapy (ACT) or cognitive behavioural therapy (CBT) [6]. The interdisciplinary approach in pain rehabilitation reflects the complexity of the chronic pain disorders seen in this group of patients, and a biopsychosocial approach considering physical, psychological, social and contextual factors is commonly applied when assessing and managing chronic pain conditions [7,8,9]. The complexity of the disorder also raises the question of which parameters one can expect to be affected by a rehabilitation programme. A recent study found that pain intensity was one of the factors with the least probability of improvement, whilst depression was the factor with the highest probability of improvement in patients after participating in an IPRP in Sweden [10].

Almost half of the patients at specialist interdisciplinary treatment clinics in Sweden can be expected to be on registered sick leave [11], and results from previous Swedish studies show that sickness absence generally tends to decrease over time for individuals attending an IPRP [12,13,14,15,16,17]. However, when comparing the effects on sickness absence after attending IPRP with other/no treatment, the studies are less conclusive, showing both advantageous effects of IPRP on sickness absence [14,15] as well as no difference to other/no treatments [12,16,17]. Several studies have shown that interventions at the workplace are important for improving the employee’s workability and for a successful RTW process [18,19,20,21,22], and including a work accommodation component in interventions has shown to be more effective for reduced sickness absence and successful RTW among patients with chronic pain than including CBT only [18]. Further, an early workplace dialogue, in addition to physiotherapy, has been shown to improve work ability among chronic pain patients [20]. However, health professionals working with RTW in IPRP units in Sweden have described how they have experienced that health and RTW are part of a linear process, where health issues need to be addressed before RTW issues [23]. This priority can also be seen by the results from a recent Swedish register study showing that although most units providing IPRP include meetings with, e.g., a rehabilitation coordinator or a rehabilitation team, only 14 out of 31 investigated units reported using interventional “measures in the workplace” [6].

The process for a successful RTW after being on sick leave may involve challenges for the employee and the employer, as well as the co-workers of the sick-listed employee [24]. The collaboration and involvement of stakeholders, including the employer, are of importance for a successful and sustainable RTW [21,25,26,27]. A recent systematic review investigating which interventions are the most effective for chronic pain patients’ RTW and staying at work further highlights the importance of a tailored adaptation to the employee’s needs at the workplace [25]. One prerequisite for achieving good workability is a good balance between demands at work and the individual’s resources, including functional capabilities and motivation [28,29]. The Demand and Ability Protocol (DAP) is a protocol used in three-party meetings for a dialogue involving the employee, the employer and a representative from the rehabilitation team/health care. The DAP considers work demands and the individual’s resources. During the structured dialogue, the employee and employer are to assess, reason and agree upon the work demands in relation to the employee’s abilities [30,31]. The aim of this study is to investigate the inclusion of an intervention using the DAP in an interdisciplinary pain rehabilitation programme compared to usual care.

## 2. Materials and Methods

### 2.1. Study Design

This is a non-randomised controlled trial including patients assigned to an interdisciplinary pain rehabilitation programme (IPRP) at one out of four included rehabilitation units in Sweden. The intervention group received a rehabilitation effort targeting their work situation (DAP intervention) in addition to the usual care provided by the IPRP. The DAP was then integrated into the IPRP. The control group received the IPRP only. Data were collected at the time of the first visit to the IPRP, when finishing the rehabilitation programme and one year after the start of the rehabilitation programme. Self-reported patient-reported outcome measures (PROMS) were collected from the Swedish Quality Registry for Pain Rehabilitation (SQRP) [32]. The study protocol for the trial was registered at Clinical Trials on 15 October 2021 with the registration number NCT05080062.

Ethical approval was obtained from the Regional Ethical Review Board (D-nr 2019-01755 and 2020-00015).

### 2.2. Participants

The participants of this study (for intervention and control) were recruited among the patients assigned to an IPRP at one of the four included units between 2017 and 2021. For a patient to be assigned to an IPRP programme, an interdisciplinary assessment team needed to determine if the patient was suitable, and they needed to fulfil the inclusion criteria: (1) they should be adults (18–65 years old), (2) have disabling chronic pain (experiencing major interference in daily life or being on sick leave, duration of pain for at least 3 months), (3) there should be no need for further medical investigations and (4) the patient should not have any ongoing major somatic or psychiatric disease, a history of significant substance abuse, or be in a state of acute crisis.

The inclusion criteria for this study were: being assigned to an IPRP programme, employed and not on full-time sick leave for more than 6 months before the start of rehabilitation.

Recruitment to the DAP intervention was made among patients who started their IPRP at one of the four included IPRP units between 2019 and 2021 and who fulfilled the inclusion criteria. All patients signed an informed consent for participation before entering the study.

Historical controls were recruited from the Swedish Quality Registry for Pain Rehabilitation (SQRP) among individuals who had been patients at the IPRP between February 2017 and September 2018. The SQRP is a register that is based on questionnaire data from patients at any of the pain rehabilitation clinics in Sweden [32]. The controls were selected by type of pain rehabilitation (they should have entered the same IPRP as the intervention group) at the same clinics. The same criteria for inclusion were then applied when assessing the control group as the intervention group. The participants of the control group signed an informed consent form at the time of assignment to the IPRP.

### 2.3. Interventions

Both the intervention group and the control group received usual care in accordance with the IPRP. The intervention group received an intervention based on the Demand and Ability Protocol (DAP) in addition to usual care.

#### 2.3.1. Usual Care—Interdisciplinary Pain Rehabilitation Programme (IPRP)

Both groups received usual care according to the 5–6-week IPRP at one of the four outpatient clinics. The programme is held in groups of 6 to 8 patients in each group. The IPRP included information and training in coping strategies, information on chronic pain and psychological and bodily reactions to chronic pain, relaxation and body-awareness training and physical and occupational therapy. The patients also had individual schedules.

#### 2.3.2. The Demand and Ability Protocol (DAP) Intervention

The patients in the intervention group participated in a three-party meeting that included the patient, his/her immediate manager and a representative from the rehabilitation team (e.g., in this study, an occupational therapist). The meeting was structured according to the DAP. The DAP intervention was integrated within the IPRP and was held when the patient had been in the programme for a few weeks. The DAP is an intervention based on the Dutch Functional Ability List and knowledge about disability in working life and is linked to the International Classification of Functioning, Disability and Health (ICF) [33]. The purpose of the DAP intervention was to let the patient and manager assess the patient’s workability in relation to the demands at the workplace in order to be able to make adequate adjustments at the workplace and promote a successful RTW. The DAP (also called “the Requirements and Functional Schedule”) was developed in Norway. This structured dialogue has primarily been used within occupational healthcare settings for assessments of an employee’s functional abilities in relation to his/her requirements at work [30,31]. The DAP is structured into six domains, assessing: (1) mental and cognitive ability, (2) basic skills and social ability, (3) tolerance for physical conditions, (4) ability to work dynamically, (5) ability to work statically and (6) ability to work certain times. Around each domain, the balance between the patient’s abilities and the demands at work is assessed by detailed questions (Table 1). The patient and the employer should rate the abilities and demands at work on a scale of 1–3 under the agreement. Areas where there is an imbalance between ability and demands can then be identified and act as a base for further actions and workplace adjustments. The protocol ends with a summary of the actions planned in relation to the patient’s RTW.

In most cases, the DAP meetings took place at the rehabilitation clinic. However, due to recommendations in conjunction with the Covid-19 pandemic, some meetings were held digitally. The patients initiated the DAP by contacting the manager for participation in the DAP. The active involvement and engagement of the patient in this process are on purpose so that the patient plays an active part in his/her rehabilitation. The DAP intervention included only one meeting, without any follow-up meetings. The meeting was led by occupational therapists who had received training in performing the DAP, in accordance with what is recommended for using the DAP.

### 2.4. Outcome Measures

Information on patient-reported outcome measures (PROMs) for each patient (in the intervention group and the control group) was retrieved from the SQRP. The patients filled in the questionnaire at the time of assessment (first visit to the clinic), immediately after finishing the IPRP and after 1 year. The outcome measurements of interest in this study were sickness absence, workability, anxiety, depression, sleep, perceived health status, life satisfaction, satisfaction with occupation and work significance.

*Sickness absence* (0–100%) was assessed by self-reported information from patients.

*Workability* was assessed by one item from the Work Ability Index (WAI) [34], where the respondent rated their current workability compared with their lifetime best on a scale ranging from 0 (completely unable to work) to 10 (workability at its best) [35].

*Anxiety and Depression* were assessed by the Hospital Anxiety and Depression scale (HAD) [36]. The two dimensions are assessed with 7 items, respectively, generating an index score for each dimension ranging from 0 (lowest risk) to 21 (highest risk). A score of 0–7 indicates “low risk”, 8–10 indicates “risk”, and 11–21 indicates “probable risk” (for anxiety or depression, respectively) [37].

*Sleep* was assessed by the Insomnia Severity Index (ISI), which comprises an index of 7 items, with a total score ranging from 0 (no problems with insomnia) to 28 (severe problems) [38,39]. The total score from the index can be categorised into: 0–7 “No clinically significant insomnia”; 8–14 “Subthreshold insomnia”; 15–21 “Clinical insomnia (moderate)”; 22–28 “Clinical insomnia (severe)” [40].

*Perceived health status* was assessed by EQ5D VAS and the EQ5D index [41,42]. The EQ5D is divided into two parts, where the first part assesses five dimensions—mobility, self-care, usual activities, pain/discomfort and anxiety/depression. Based on these five dimensions, an index can be calculated (the EQ5D index), where a lower score indicates the worst health status and a higher score indicates better health. The index normally ranges between 0–1, although negative values may occur [42]. The second part of the questionnaire assesses health status on a vertical VAS scale, ranging from 0–100, where 0 represents the worst imaginable health and 100 represents the best imaginable health [42].

*Life satisfaction* was assessed by two items from the Life Satisfaction checklist (LiSAT-11): Satisfaction with life as a whole and satisfaction with a vocational situation [43,44]. Satisfaction for each domain respectively was scored on a 6-level Likert scale, ranging from 1 “Very dissatisfied” to 6 “Very satisfied”.

*Importance of work* was assessed by the single item “What importance does work have for you, apart from being a source of income?” The item is responded to using a five-level Likert scale, ranging from 1 “Very high importance” to 5 “No importance”.

Self-reported information regarding physical activity, BMI and type of work was further assessed for descriptive purposes.

### 2.5. Data Analyses

Data were analysed using the computer program SPSS 28.0 (Armonk, NY, USA: IBM Corp. 2016). Due to non-distributed data, non-parametric tests were applied in all data analyses, and descriptive data are presented by medians and interquartile ranges (IQRs).

For analyses of differences between groups, the Mann–Whitney U-test was used. When analysing changes in outcome measures between baseline and post-rehabilitation and one-year follow-up, respectively, the Wilcoxon signed rank test was used.

For analyses of differences in changes between the DAP intervention group and the control group, variables were created to represent a change in each outcome measure between baseline and post-rehabilitation and baseline and one-year follow-up, respectively. Thereafter, differences (in changes) between the groups could then be calculated using a Mann–Whitney U-test. Statistical significance was set to *p* < 0.05.

## 3. Results

The participants in the DAP intervention group were between 22 and 61 years old, and 83% were women. The participants in the control group were between 23 and 62 years old, where 91% were women. There was no difference between the DAP intervention group and the control group at baseline regarding any of the baseline characteristics or the outcome variables of interest for this study (See Table 2). There was a variation in time between inclusion in the study (baseline measurement at the first visit to the clinic) and actually starting the rehabilitation programme (median 26.5 weeks (range 4–78) in the control group and median 26.5 weeks (range 9–81) in the DAP intervention group). The difference between the groups was not statistically significant (*p* = 0.687).

### 3.1. Changes in Outcome Measures after Pain Rehabilitation

Analysing the intervention group and the control group together, statistically significant positive effects were seen on depression, EQ5D VAS and sleep after finishing the pain rehabilitation programme (post-rehabilitation). At the one-year follow-up, positive effects were seen on most outcome measures—sickness absence, importance of work, anxiety, depression, EQ5D VAS, the EQ5D index and sleep when compared to baseline measurements. However, no effects were seen on workability, life satisfaction or satisfaction with vocation (Table 3).

### 3.2. Effects of including DAP in the Interdisciplinary Rehabilitation Programme

Positive effects were seen on depression in both the intervention group and the control group after finishing the rehabilitation (post-rehab). In addition, positive effects were seen on sleep in the intervention group at the post-rehabilitation follow-up.

The positive effect on depression remained at the one-year follow-up in both the intervention group and the control group, and the positive effect on sleep remained in the intervention group. In addition, effects were seen on anxiety in the intervention group but not in the control group. Effects on the importance of work and EQ5D VAS were seen in both groups, although they did not quite reach statistical significance in the intervention group. In addition, effects were seen on sickness absence and the EQ5D index in the control group. These effects were, however, not seen in the intervention group.

No differences were seen when comparing the changes between baseline and post-rehab and baseline and one-year follow-up in the intervention group and the control group (Table 4).

At one-year follow-up, 10 out of 18 (56%) were on sick leave in the intervention group, and 4 out of 18 (22%) were on sick leave among the controls. The change in prevalence of sick leave between baseline and one-year follow-up was not significant in the intervention group (*p* = 1.000) or control group (0.250). However, information on sickness absence was missing for 12 patients in the intervention group and 16 patients in the control group.

## 4. Discussion

This study has investigated the inclusion of an intervention based on the DAP in an IPRP, compared to usual care with IPRP only. In short, no differences were seen in effect between the group that received the additional DAP intervention and the group that received usual care. The purpose of DAP is to promote a successful RTW and sustainable work situation, and the effects of the intervention were evaluated by analysing changes in factors directly related to the work situation—sickness absence, workability, satisfaction with vocational situation and the importance of work—as well as factors related to health–anxiety, depression, EQ5D, sleep and satisfaction with life.

Both groups showed an overall tendency for improvement in the PROMs, namely, self-reported anxiety, depression and EQ5D, especially at the one-year follow-up, although some changes did not reach statistical significance. The effect of IPRP on these PROMs is in line with previous studies evaluating the effect of IPRP [10,45]. The patients who received the DAP intervention showed significant improvements in sleep, both at the post-rehabilitation follow-up and at the one-year follow-up. This change was not seen in the control group. However, the difference in change between the intervention group and the control group was not significant, which may be a result of low statistical power. The changes in sleep seen among the patients receiving the DAP intervention are, however, worth considering since sleep has shown to be a valuable predictor for pain disorders [9,46] and sickness absence [47] and may reflect several important aspects of changes in health among individuals with chronic pain [46]. Further, satisfactory sleep (and possibly also improved sleep quality) has been suggested to be associated with the improvement of chronic pain conditions [48,49].

Regarding the factors more specifically related to the work situation, the results are less clear. A tendency of increased workability was seen at the post-rehabilitation follow-up in the intervention group but not in the control group. Although the change in workability was not statistically significant (*p* = 0.085), the increase in the median from 5.0 to 6.0 may indicate an overall improvement in workability. However, no effects could be seen in the one-year follow-up in any of the groups. Further, in the present study, significant changes in sickness absence were seen between baseline and one-year follow-up in the control group but not in the intervention group. Regarding the results on sickness absence in the present study, there are a couple of aspects one should consider when interpreting the results. Firstly, information on sickness absence was missing for many patients in both groups. Secondly, at baseline, 55% of the intervention group were on sick leave compared to 35% of the control group. This difference (although not statistically significant) may indicate that the patients who agreed to participate in the DAP intervention may represent a different group of patients regarding work situations and/or disease severity. Further, although the inclusion criteria allowed for only a maximum of six months of full-time sick leave before inclusion, some patients may have been on part-time sick leave for much longer. Among the patients in the intervention group, 55% had a type of work that they reported to be medium heavy or heavy, whereas, in the control group, only 22% reported a type of work that was medium heavy or heavy. Given the presence of a chronic pain condition, the type of work could be highly relevant for the need for sickness absence.

Both groups rated their work as more important at the one-year follow-up than at baseline, although the change did not quite reach statistical significance in the intervention group. As for satisfaction with vocation, an improvement was seen only in the control group and only during the follow-up post-rehabilitation. No effect was seen on life satisfaction in either of the groups. In general, both groups rate their life satisfaction and satisfaction with vocation lower than the normative median in Sweden (norm value = 5) [50].

The (lack of) findings regarding factors more specifically related to the work situation were not expected since the intervention group received the DAP intervention, supposedly targeting a successful RTW process. The DAP intervention actively involves the manager of the individual with chronic pain in the RTW process [31], and a good relationship and collaboration between the manager and the employee has previously been highlighted as important for a successful RTW [25,51,52,53,54,55,56,57]. In two recent studies of ours, investigating the patient’s perspective [51] and the manager’s perspective (not yet published) when participating in the DAP, we found that the intervention was perceived to promote their relationship and collaboration. The participants in these two qualitative studies (in particular, the managers) further expressed that they believed the DAP intervention would benefit from including more than one meeting to follow up on the RTW process. More continuous contact and active involvement of the manager are supported by a review by Durand et al., where six steps are suggested for managing work absence due to musculoskeletal disorders and a successful RTW process: (1) time off and recovery period; (2) initial contact with the worker; (3) evaluation of the worker and his/her job tasks; (4) development of a return-to-work plan with accommodations; (5) work resumption; (6) follow-up on the return-to-work process.

Previous studies have highlighted the importance of involving the workplace in the rehabilitation process, even before the worker has made a full recovery [23,58], and suggested the benefits of continuous supervision of vocational issues [23]. A rehabilitation programme may further benefit from including features with a concrete focus on vocational issues [23], e.g., by specified goals achieved based on personalised target activities [59] for RTW and the early identification of factors that hinder RTW. The RTW is an outcome of relevance after participating in an IPRP, and the programme could be better tailored to address the reciprocal relationship between work and health [23]. Previous studies evaluating the effect of IPRP have shown unclear results on workability and sickness absence [12,16,17], which may suggest that the involvement of the employer or other stakeholders of relevance in the rehabilitation is of importance for the patient’s RTW process. The results from our study, however, did not show any clear effects on workability or sickness absence when doing so. In future studies involving the employer in the rehabilitation, it may be beneficial to involve the employer earlier in the process and to plan for more than one meeting (e.g., structured by the DAP) during the RTW process. The DAP also has the potential to support the employer and employee in reaching an agreement on which accommodations may not be necessary, which would facilitate directing resources to issues with a higher potential of having an impact. Furthermore, it may be of value to review relevant outcomes for the evaluation of the RTW process. In this study, we chose to evaluate the effect of the DAP intervention based on outcome measures reflecting health in a more global meaning, as well as factors more specifically related to the work situation. To the best of our knowledge, there is no consensus around a core outcome set when evaluating interventions targeting RTW and sustainable work situations. A recently published review has initiated the work of establishing a core outcome set for the evaluation of work participation by suggesting categorisation into “employment status”, “absence from work”, “at-work productivity loss” and “employability”. The study, however, concludes that there is a large variability in how outcomes related to these four categories are measured and that more work is needed to establish a core outcome [60]. To evaluate the change in sickness absence may appear to be a sound choice; however, sickness absence may not solely reflect poor workability or changes in health. This discrepancy was seen in this study, e.g., by looking at the absence of change in workability in the control group, although a change in sickness absence was seen.

### Limitations

One limitation of this study is the low number of patients recruited, which resulted in a lack of statistical power and limited possibilities to detect effects.

Another limitation was that the study was non-randomised. Furthermore, the intervention group and the control group were recruited at two different periods of time. The intervention group started their rehabilitation programme between 2019 and 2021. It is quite possible that the outbreak of COVID-19 in March 2020 and onward had an impact on health, the work situation, and the performance of the pain rehabilitation programme. These issues make the comparability of the intervention group and the control group less robust. Furthermore, this study does not include an outcome measure regarding pain since the DAP is not a pain intervention and the focus is on measuring functioning, work ability and life quality. Still, it could have been interesting to also examine the symptoms in relation to the interventions.

## 5. Conclusions

The results from this study showed that the IPRP had an effect on the PROMs of self-reported anxiety, depression and EQ5D. The effect was clearer at the one-year follow-up. Adding the DAP intervention to usual care within the IPRP seems to have the potential to improve sleep among the patients, which may indicate an overall improvement regarding health outcomes from a longer perspective. The results were less clear, however, regarding the work-related outcomes of sickness absence and workability. No significant differences in effects were seen between the group receiving both usual care (IPRP) and the DAP intervention and the group receiving usual care only for any of the included outcomes.

## Figures and Tables

**Table 1 ijerph-19-16614-t001:** A presentation of the six domains included in the DAP, the number of items in each domain, and examples of items *.

Domains	Number of Items in Domain	Examples of Items in Domain
1. Mental and cognitive ability	7	Concentration, memory, acting goal-oriented and independent
2. Basic skills and social ability	10	Writing, reading, handling conflicts and own emotions
3. Tolerance for physical conditions	8	Heat, cold, personal protective equipment, dust, vibrations
4. Ability to work dynamically	14	Work with hand and fingers, forward bending, rotation of body
5. Ability to work statically	6	Sit, stand, work with arms above shoulders or in a forward bent position
6. Ability to work at certain times	3	Working hours per day or week

* The content of the table has been reproduced from a previous study, with permission from the authors [31].

**Table 2 ijerph-19-16614-t002:** Characteristics of the participants in the intervention group and the control group.

		DAP Intervention *N* = 30		Control Group *N* = 34		*p*-Value ^a^
** *Baseline descriptive* **		*n*		*n*		
Age, median (IQR)		30	43 (36–52)	34	47 (38–52)	0.322
Gender, (female) *n* (%)		30	25 (83)	34	31 (91)	0.344
BMI, median (IQR)		24	28.5 (24.1–36.6)	30	27.2 (22.4–29.9)	0.166
Type of work, *n* (%)		24		28		0.086
	Administrative		8 (33)		16 (54)	
	Light repetitive		4 (17)		7 (25)	
	Medium heavy		9 (38)		2 (7)	
	Heavy repetitive		3 (13)		3 (11)	
	Handling heavy materials		0 (0)		1 (4)	
Physical training, *n* (%)		28		31		0.564
	0 min		10 (36)		14 (42)	
	<30 min		3 (11)		8 (24)	
	30–60 min		5 (18)		5 (15)	
	60–90 min		3 (11)		2 (6)	
	90–120 min		3 (11)		1 (3)	
	>120 min		4 (14)		3 (9)	
Physical activity, exercise, *n* (%)		28		31		0.504
	0 min		0 (0)		1 (3)	
	<30 min		4 (14)		2 (6)	
	30–60 min		4 (14)		7 (21)	
	60–90 min		3 (11)		5 (15)	
	90–150 min		5 (18)		3 (9)	
	150–300 min		4 (14)		9 (27)	
	>300 min		8 (29)		6 (18)	
** *Outcome measures* **						
Sickness absence (yes), *n* (%)		22	12 (55)	29	10 (35)	0.152
Sickness absence (0–100%), median (IQR)		22	19.0 (0.0–50.0)	29	0.0 (0.0–40.0)	0.219
Workability, median (IQR)		25	5.0 (3.0–7.0)	32	5.5 (3.0–7.8)	0.432
Importance of work, *n* (%)		28		34		0.396
	Very high		16 (57)		14 (41)	
	High		10 (36)		13 (38)	
	Some		2 (7)		6 (18)	
	Hardly any		0 (0)		1 (3)	
	No importance		0 (0)		0 (0)	
HAD anxiety sum, median (IQR)		28	8.5 (4.5–12.8)	33	10.0 (5.0–13.0)	0.873
HAD anxiety, *n* (%)		28		33		0.782
	Small risk		12 (43)		13 (39)	
	Risk		4 (14)		7 (21)	
	Probable risk		12 (43)		13 (39)	
HAD depression, sum, median (range)		28	8.5 (6.0–12.0)	33	7.0 (4.0–10.0)	0.214
HAD depression, *n* (%)		28		33		0.180
	Small risk		12 (43)		18 (55)	
	Risk		5 (18)		9 (27)	
	Probable risk		11 (39)		6 (18)	
EQ5D, VAS; median (IQR)		26	45.0 (31.0–72.5)	32	40.0 (30.0–53.8)	0.151
EQ5D index; median (IQR)		26	0.09 (−0.08–0.64)	34	0.18 (0.09–0.66)	0.424
Satisfaction with life, median (IQR)		27	4.0 (2.0–5.0)	28	4.0 (4.0–5.0)	0.243
Satisfaction with vocation, median (IQR)		27	4.0 (2.0–5.0)	28	4.0 (2.0–5.0)	0.756
Sleep (ISI), sum, median (IQR)		27	17.0 (11.0–21.0)	31	16.0 (10.0–22.0)	0.761
Sleep (ISI), *n* (%)		27		31		0.666
	No clinically significant insomnia		4 (15)		2 (7)	
	Subthreshold insomnia		7 (26)		11 (36)	
	Clinical insomnia (moderate)		10 (37)		10 (32)	
	Clinical insomnia (severe)		6 (22)		8 (26)	

^a^ Differences analysed by Mann–Whitney test when continuous, chi-squared when categorical. *WAI*—higher score indicates better workability; *Importance of work*—higher score indicates lower importance; *Anxiety and depression* (*HAD*)—higher score indicates worse problems; *EQ5D*—higher score indicates better health; *Sleep* (*ISI*)—higher score indicates worse problems; *Satisfaction with Life and Satisfaction with Occupation* (*LiSAT*)—higher score indicates more (better) satisfaction.

**Table 3 ijerph-19-16614-t003:** Effects of participating in the interdisciplinary pain rehabilitation programme. Differences between baseline and post-rehab and baseline and one-year follow-up were analysed by Wilcoxon signed rank test.

	Baseline		Post-Rehabilitation		Difference	One-Year Follow-Up		Difference
	*N*	Median (IQR)	*N*	Median (IQR)	*p*-value	*N*	Median (IQR)	*p*-value
Sickness absence	51	0.0 (0.0–50.0)	-	-	-	36	0.0 (0.0–25.0)	0.018 *
Workability (WAI)	57	5.0 (3.0–7.0)	55	6.0 (3.0–7.0)	0.426	52	6.0 (3.0–8.0)	0.510
Importance of work	62	2.0 (1.0–2.0)	55	2.0 (1.0–2.0)	0.159	51	2.0 (1.0–3.0)	0.006 *
Anxiety (HAD)	61	9.0 (5.0–13.0)	55	8.0 (5.0–10.0)	0.134	53	7.0 (5.0–12.0)	0.010 *
Depression (HAD)	61	8.0 (5.0–11.0)	54	6.0 (3.0–8.0)	<0.001 *	53	6.0 (4.0–8.0)	0.003 *
EQ5D VAS	60	40.0 (30.0–60.0)	53	50.0 (35.0–60.0)	0.049 *	52	57.5 (40.0–70.0)	0.001 *
EQ5D Index	60	0.16 (−0.02–0.65)	55	0.23 (0.09–0.69)	0.051	52	0.66 (0.13–0.73)	<0.001 *
Sleep (ISI)	58	16.0 (11.0–21.25)	55	14.0 (8.0–19.0)	0.026 *	45	14.0 (7.0–18.75)	0.012 *
Satisfaction with life	55	4.0 (3.0–5.0)	45	4.0 (4.0–5.0)	0.315	45	4.0(4.0–5.0)	0.870
Satisfaction with vocation	55	4.0 (2.0–5.0)	46	3.0 (2.0–5.0)	0.090	52	4.0(3.0–5.0)	0.970

* *p* < 0.05; *WAI*—higher score indicates better workability; *Importance of work*—higher score indicates lower importance; *Anxiety and depression* (*HAD*)—higher score indicates worse problems; *EQ5D*—higher score indicates better health; *Sleep* (*ISI*)—higher score indicates worse problems; *Satisfaction with Life and Satisfaction with Occupation* (*LiSAT*)—higher score indicates more (better) satisfaction.

**Table 4 ijerph-19-16614-t004:** Results from the analyses of changes in outcomes after DAP post rehabilitation and at 1-year follow-up in the intervention group and control group, respectively. The right column presents differences between the intervention group and the control group in changes (baseline–post rehab, and baseline-1 year follow-up). Differences between baseline and post rehab or baseline and 12-month follow-up in the DAP-intervention group and control group respectively, were analysed by Wilcoxon signed rank test.

	DAP Intervention			Control			Diff. Baseline–Post-Rehab/ Baseline–One-Year Follow-Up		
							DAP	Control	Diff
Outcomes	*N*	Median (IQR)	*p* ^a^	*N*	Median (IQR)	*p* ^a^	Median (IQR)	Median (IQR)	*p* ^b^
** *Sickness absence* **									
Baseline	22	19.0 (0.0–50.0)		29	0.0 (0.0–40.0)				
Post-rehab							-	-	
1 year	18	20.0 (0.0–100.0)	0.156	18	0.0 (0.0–50.0)	0.042 *	0.0 (−31.2–5.0)	0.0 (−23.2–0.0)	0.759
***Workability* (*WAI*) **									
Baseline	25	5.0 (3.0–7.0)		32	5.5 (3.0–7.8)				
Post-rehab	24	6.0 (4.25–7.0)	0.085	31	5.0 (3.0–7.0)	0.522	0.0 (−1.0–2.0)	0.0 (−1.0–1.0)	0.127
1 year	18	6.0 (2.75–8.0)	0.944	34	5.5 (3.75–8.0)	0.414	−1.0 (−2.0–2.0)	0.0 (−1.0–2.7)	0.489
** *Importance of work* **									
Baseline	28	1.0 (1.0–2.0)		34	2.0 (1.0–2.0)				
Post-rehab	24	1.5 (1.0–2.0)	0.132	31	2.0 (1.0–3.0)	0.558	0.0 (0.0–1.0)	0.0 (0.0–0.0)	0.220
1 year	18	2.0 (1.0–2.0)	0.058	33	2.0 (1.0–3.0)	0.046 *	0.0 (0.0–1.0)	0.0 (0.0–0.5)	0.440
***Anxiety* (*HAD*) **									
Baseline	28	8.5 (4.5–12.8)		33	10.0 (5.0–13.0)				
Post-rehab	24	6.5 (5.0–9.75)	0.302	31	8.0 (5.0–11.0)	0.287	−1.0 (−3.0–2.0)	−1.0 (−3.0–1.2)	0.822
1 year	19	7.0 (5.0–10.0)	0.018 *	34	7.0 (4.0–12.0)	0.134	−2.0 (−4.0–1.0)	−1.0 (−3.5–1.5)	0.405
***Depression* (*HAD*) **									
Baseline	28	8.5 (6.0–12.0)		33	7.0 (4.0–10.0)				
Post-rehab	24	7.0 (3.0–8.0)	0.006 *	30	6.0 (3.8–7.5)	0.003 *	−2.0 (−5.0–1.0)	−2.0 (−3.5–0.0)	0.354
1 year	19	7.0 (5.0–8.0)	0.013 *	34	6.0 (3.0–8.3)	0.049 *	−3.0 (−4.0–1.0)	−1.0 (−3.5–0.0)	0.322
** *EQ5D VAS* **									
Baseline	26	45.0 (31.0–72.5)		32	40.0 (30.0–53.8)				
Post-rehab	24	57.5 (49.3–64.8)	0.393	29	40.0 (35.0–60.0)	0.058	1.0 (−10.0–20.0)	5.0 (−3.7–14.5)	0.453
1 year	19	60.0 (30.0–75.0)	0.054	33	52.0 (40.0–67.0)	0.006 *	5.0 (−2.5–18.0)	10.0 (0.0–27.0)	0.681
**EQ5D Index**									
Baseline	26	0.09 (−0.08–0.64)		34	0.18 (0.09–0.66)				
Post-rehab	24	0.24 (0.03–0.62)	0.289	31	0.20 (0.09–0.69)	0.083	0.1 (−0.1–0.3)	0.0 (−0.0–0.2)	0.820
1 year	19	0.62 (0.09–0.73)	0.093	33	0.66 (0.16–0.73)	0.002 *	0.1 (0.0–0.5)	0.1 (0.0–0.4)	0.814
***Sleep* (*ISI*) **									
Baseline	27	17.0 (11.0–21.0)		31	16.0 (10.0–22.0)				
Post-rehab	24	12.5 (7.3–16.0)	0.019 *	31	17.0 (9.0–21.0)	0.415	−1.0 (−7.0–2.0)	−1.0 (−4.7–2.0)	0.193
1 year	19	12.0 (6.0–16.0)	0.008 *	33	16.0 (7.5–19.5)	0.269	−3.0 (−6.0–−1.5)	−1.0 (−5.0–2.0)	0.151
** *Satisfaction with life* **									
Baseline	27	4.0 (2.0–5.0)		28	4.0 (4.0–5.0)				
Post-rehab	19	4.0 (3.0–5.0)	0.133	26	4.0 (4.0–4.25)	0.805	0.0 (−1.0–1.0)	0.0 (−0.5–0.5)	0.159
1 year	19	4.0 (4.0–5.0)	0.248	26	4.0 (4.0–5.0)	0.417	−1.0 (0.0–1.0)	0.0 (0.0–0.0)	0.271
** *Satisfaction with vocation* **									
Baseline	27	4.0 (2.0–5.0)		28	4.0 (2.0–5.0)				
Post-rehab	19	3.0 (3.0–4.0)	0.659	27	3.0 (1.0–5.0)	0.059	0.0 (−1.0–1.0)	0.0 (−1.0–0.0)	0.364
1 year	19	4.0 (3.0–5.0)	0.659	26	4.0 (2.0–5.0)	0.651	0.5 (−0.7–1.0)	0.0 (−1.0–1.0)	0.347

* *p* < 0.05; ^a^ analysed by Wilcoxon signed rank test; ^b^ difference between the DAP intervention group and the control group in median changes, analysed by the Mann–Whitney U-test. *WAI*—higher score indicates better workability; *Importance of work*—higher score indicates lower importance; *Anxiety and depression* (*HAD*)—higher score indicates worse problems; *EQ5D*—higher score indicates better health; *Sleep* (*ISI*)—higher score indicates worse problems; *Satisfaction with Life and Satisfaction with Occupation* (*LiSAT*)—higher score indicates more (better) satisfaction.

## Data Availability

The data presented in this study are available on request from the corresponding author. The data are not publicly available due to ethical restrictions.

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
