# Peer review of "Including a Three-Party Meeting Using the Demand and Ability Protocol in an Interdisciplinary Pain Rehabilitation Programme for a Successful Return to Work Process"

_ijerph, 2022, doi:10.3390/ijerph192416614_

Round 1
Reviewer 1 Report
Thank you for sending this manuscript. It is a well-written mansucript with an interesting topic. I have no feedback on how to improve.
Author Response
Thank you for your kind words regarding our manuscript. We have thoroughly gone through the whole manuscript once again to check the language and minor changes have been done.
Reviewer 2 Report
The aim of this study was to investigate the inclusion of an intervention using the DAP in an interdisciplinary pain rehabilitation programme (IPRP), compared to usual care. This non-randomised controlled trial included patients assigned to an IPRP in Sweden. The intervention group received a DAP intervention targeting their work situation in addition to the usual care provided by the IPRP.
In Sweden, musculoskeletal disorder is the second most common cause of sickness absence. The participants in the DAP intervention group were between 22 and 61 years old, and 83% were women. The participants in the control group were between 23 and 62 years old, where 91% were women. Analysing the intervention group and control group together, statistically significant positive effects were seen on depression. However, no effects were seen on workability, life satisfaction or satisfaction with vocation.
No differences were seen in effect between the group that received the additional DAP intervention and the group that received usual care. Both groups showed an overall tendency of improvements in the PROMs self-reported anxiety, depression and EQ5D, especially at the 1-year follow-up, although some changes did not reach statistical significance. The patients who received the DAP intervention showed significant improvements in sleep, both at the post rehabilitation follow-up and at the 1-year follow-up.
Regarding the factors more specifically related to the work situation, the results are less clear. However, no effects could be seen in the 1-year follow-up in any of the groups.
Regarding the results on sickness absence in the present study, information on sickness absence was missing for many patients in both groups. at baseline, 55% in the intervention group were on sick leave compared to 35% in the control group. This difference (although not statistically significant) may indicate that the patients who agreed to participate in the DAP intervention may represent a different group of patients regarding work situation and/or disease severity. Among the patients in the intervention group, 55% had a type of work that they reported to be medium heavy or heavy, whereas in the control group, only 22% reported a type of work that was medium heavy or heavy.
Both groups rated their work as more important at the 1-year follow-up than at baseline, although the change did not quite reach statistical significance in the intervention group.
This studies highlight the importance of involving the workplace in the rehabilitation process, even before the worker has made full recovery. The study, however, concludes that there is a large variability in how outcomes related to these four categories are measured, and that more work is needed for establishing a core outcome.
Conclusion:
One limitation of this study is the low number of patients recruited, which resulted in a lack of statistical power and limited possibilities to detect effects and limitation is that the intervention group and the control group were recruited at two different periods of time. It is quite possible that the outbreak of Covid-19 in March 2020 and onward had an impact on health, the work situation, as well as the performance of the pain rehabilitation programme.
Author Response
Thank you for taking the time to review our manuscript.
Reviewer 3 Report
Dear authors,
I would like to thank you for the opportunity to review your manuscript. This research aims to examine the use of DAP in a pain rehabilitation programme.
## General comments to authors: This non-randomized controlled trial is well-written and presents an interesting research topic that could be translated to clinical practice. However, before considering for publication manuscript should address some minor issues. I hope my comments could help authors to improve their manuscript.
Abstract:
# Comment 1: Results from quantitative analysis have to appeared in the abstract.
# Comment 2: Conclusion in the abstract should be in line with the conclusion of the manuscript.
Methods:
# Comment 1: Section 2.5. Software and significance of p-value have to be specified.
# Comment 2: Considering one of the focuses of the research is pain rehabilitation programme, why a specific outcome for pain was not included?
Discussion:
# Comment 1: The limitations section should be expanded, because this manuscript has more limitations than the mentioned (e.g., the lack of randomization).
Author Response
Thank you for taking the time to review our manuscript. We have revised the manuscript according to your comments. All changes are marked with track changes.
Abstract
Comment 1: Results from quantitative analysis have to appeared in the abstract.
Answer 1: Thank you for an important reflection, the results have been made clearer in the abstract.
Comment 2: Conclusion in the abstract should be in line with the conclusion of the manuscript.
Answer 2: Thank you for the comment, the conclusion of the abstract have been modified. The new text is changed as follows:
“In conclusion, no significant differences in effects were seen between the group receiving IPRP or DAP intervention. Adding the DAP intervention to IPRP seemed to have the potential to improve sleep among the patients, which may indicate an overall improvement regarding health outcomes from a longer perspective. The results were less clear, however, regarding the work-related outcomes of sickness absence and workability.”
Methods
Comment 1: Section 2.5. Software and significance of p-value have to be specified.
Answer 1: Thank you for an important observation, a sentence about significance of p-value and software have been added in section 2.5.
“Statistical significance was set to p < 0.05. Statistical analysis was conducted with SPSS version 28.0 and that has been added in the manuscript.
Comment 2: Considering one of the focuses of the research is pain rehabilitation programme, why a specific outcome for pain was not included?
Answer 2: This is an interesting consideration. When designing the study we focused on including outcomes measuring functioning and life quality, since DAP is not a pain intervention. Furthermore, there is a question about pain in EQ5D. A comment regarding this issue have been added in the limitation section.
Discussion
Comment 1: The limitations section should be expanded, because this manuscript has more limitations than the mentioned (e.g., the lack of randomization).
Answer 1: Thank you for an important comment. The limitation (lack of randomization) has been added to the limitation section.
“Another limitation was that the study was non-randomized, furthermore…”. We have also added a comment on the lack of pain as an outcome measure in the study.